# Real-time observation of tetrapyrrole binding to an engineered bacterial phytochrome

Yusaku Hontani [1,6], Mikhail Baloban [2], Francisco Velazquez Escobar[3], Swetta A. Jansen[1], Daria M. Shcherbakova[2], Jörn Weißenborn[1], Miroslav Kloz[1,4], Maria Andrea Mroginski [3], Vladislav V. Verkhusha [2,5] & John T. M. Kennis [1✉]

Near-infrared fluorescent proteins (NIR FPs) engineered from bacterial phytochromes are widely used for structural and functional deep-tissue imaging in vivo. To fluoresce, NIR FPs covalently bind a chromophore, such as biliverdin IXa tetrapyrrole. The efficiency of biliverdin binding directly affects the fluorescence properties, rendering understanding of its molecular mechanism of major importance. miRFP proteins constitute a family of bright monomeric NIR FPs that comprise a Per-ARNT-Sim (PAS) and cGMP-specific phosphodiesterases - Adenylyl cyclases - FhlA (GAF) domain. Here, we structurally analyze biliverdin binding to miRFPs in real time using time-resolved stimulated Raman spectroscopy and quantum mechanics/molecular mechanics (QM/MM) calculations. Biliverdin undergoes isomerization, localization to its binding pocket, and pyrrolenine nitrogen protonation in <1 min, followed by hydrogen bond rearrangement in ~2 min. The covalent attachment to a cysteine in the GAF domain was detected in 4.3 min and 19 min in miRFP670 and its C20A mutant, respectively. In miRFP670, a second C–S covalent bond formation to a cysteine in the PAS domain occurred in 14 min, providing a rigid tetrapyrrole structure with high brightness. Our findings provide insights for the rational design of NIR FPs and a novel method to assess cofactor binding to light-sensitive proteins.

[1] Department of Physics and Astronomy, Vrije Universiteit Amsterdam, Amsterdam 1081 HV, The Netherlands. [2] Departments of Anatomy and Structural Biology, Albert Einstein College of Medicine, Bronx, NY 10461, USA. [3] Institut für Chemie, Technische Universität Berlin, Sekr. PC 14, Straße des 17. Juni 135, Berlin D-10623, Germany. [4] ELI-Beamlines, Institute of Physics, Na Slovance 2, 182 21 Praha 8, Czech Republic. [5] Medicum, Faculty of Medicine, University of Helsinki, Helsinki 00290, Finland. [6] Present address: School of Applied and Engineering Physics, Cornell University, Ithaca, NY 14853, USA. ✉email: j.t.m.kennis@vu.nl

In the past two decades, the advent of genetically encoded fluorescent proteins (FPs) has revolutionized the life sciences by enabling visualization of cellular processes using fluorescence microscopy[1]. The next challenge is to image cellular and organ function in mouse and rat models, which can make a key contribution to our understanding of human health and disease. Optical imaging in mammalian tissues is difficult because of the high degree of absorption and scattering. An optical 'transparency window' exists in mammalian tissue that spans near-infrared (NIR) wavelengths between ~650–900 nm where light absorption and scattering, as well as autofluorescence of endogenous fluorophores, are minimal[2]. NIR light can probe tissue at depths of millimeters to few centimeters, enabling whole-body imaging and optogenetic control of small animals.

Bacterial phytochrome photoreceptors (BphPs) constitute a new class of molecular templates to engineer NIR FPs[3–10]. Recently, a new generation of bright monomeric NIR FPs became available, known as miRFPs[8,9]. To fluoresce, NIR FPs covalently bind a linear tetrapyrrole biliverdin-IXα (BV), the product of heme catabolism ubiquitously occurring in mammalian cells, whereas an autocatalytic chromophore maturation proceeds in FPs of the green fluorescent protein (GFP)-like family that absorbs and fluoresces in the visible range of light spectrum[11–14]. The BV binding mechanism constitutes a key issue because BV concentrations are often limiting in mammalian tissues, and other tetrapyrrole cofactors like protoporphyrins, compete for binding to NIR FP apoproteins[15]. Thus, the brightness of engineered NIR FPs in mammalian cells and tissues is a complex function of their intrinsic fluorescence properties (molecular brightness), the efficiency of BV binding and the presence of other tetrapyrroles. Understanding these processes at the molecular level may provide rational engineering of NIR FPs, resulting in NIR FPs with more efficient BV binding and consequently enhanced brightness in mammalian tissues.

To assess the BV binding, one needs a structurally sensitive time-resolved method that is sufficiently sensitive in solution. A key quality of our recently developed watermarked stimulated Raman method[16–22] is that high-quality ground-state Raman spectra can be obtained within seconds of signal averaging while being entirely insensitive to fluorescence. This means that BV binding process, ranging from minutes to tens of minutes[23], can be followed in real time after mixing apoprotein with BV. Thus, with this method, we can follow the vibrational signatures of BV intermediates as they initially bind to NIR FP apoprotein and progressively evolve to the fluorescent state. Combining the Raman technique with time-resolved absorption spectroscopy, we can study the BV binding dynamics to miRFPs[8].

Because of their small size (~35 kDa), monomeric state and high brightness in NIR, the family of miRFP proteins is widely used for deep-tissue structural and functional imaging[8,24,25]. Among them, miRFP670 is the most blue-shifted and shows the highest fluorescence quantum yield[8], thus has a high significance for multi-color NIR imaging[9]. miRFP670 has two protein domains, PAS and GAF domains, both of which have a Cys near the chromophore binding pocket (CBP) that can form a C–S covalent thioether bond to BV. Native BphPs have a Cys in the PAS domain near the CBP located in the GAF domain, so they form only a single thioether bond with the BV chromophore[26–30]. However, miRFP670 has Cys residues in both PAS and GAF domains and consequently a fraction of BV forms thioether bonds with both Cys simultaneously, while the other fraction of BV binds only to Cys in the GAF domain (Fig. 1)[31]. Two C–S bonds are unique to NIR FPs and have not been observed in BphPs nor in any other phytochrome. A similar dual C–S binding was also detected in other NIR FPs, such as dimeric iRFP670 and iRFP682, and it was proposed that the two thioether bonds increase the fluorescence quantum yield by increasing the rigidity of the bound BV adduct[32]. Here we investigate the molecular mechanism by which this unique BV binding occurs, using time-resolved absorption and stimulated Raman spectroscopies on the seconds to minutes timescale, in combination with quantum mechanical/molecular mechanical (QM/MM) calculations. We find that BV

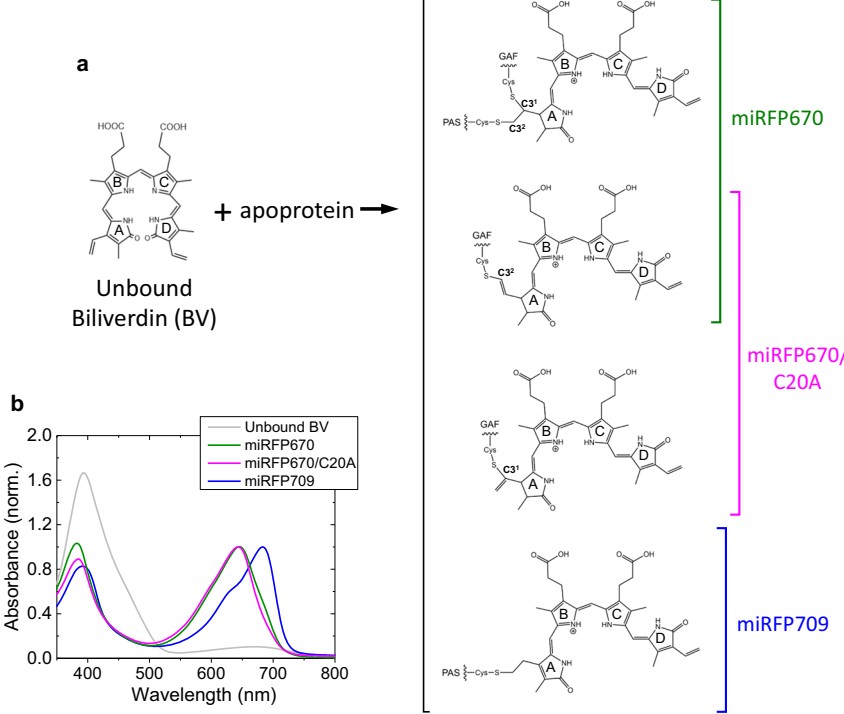

**Fig. 1 Biliverdin binding to miRFPs and absorption spectra. a** Biliverdin (BV) structures in miRFP670, miRFP670/C20A and miRFP709. The molecular structures are reproduced from Baloban et al.[31]; **b** steady-state UV–vis absorption spectra of BV in aqueous buffer solution (gray), miRFP670 (green), miRFP670/C20A (magenta) and miRFP709 (blue).

undergoes isomerization, localization to its binding pocket, and pyrrolenine nitrogen protonation in <1 min, followed by a hydrogen bond rearrangement in 2 min. The covalent attachment to a Cys in the GAF domain was detected in 4.3 min in miRFP670 and 19 min in its C20A mutant. In miRFP670, a second covalent bond formation to a Cys in the PAS domain occurred in 14 min, providing a rigid tetrapyrrole structure with high brightness. Our findings may provide insights for the rational design of NIR FPs and a novel method to assess cofactor binding to light-sensitive proteins.

## Results

**Time-resolved experiments in the miRPF670/C20A mutant.** First we studied the BV incorporation dynamics of the miRFP670/C20A mutant that has a single Cys residue in the GAF domain (Fig. 1). In BphP1-FP, which is a dimeric protein engineered from the same parental protein as miRFP670 (*Rp*BphP1)[23], C–S binding to Cys in the GAF domain both via C3[1] and C3[2] were detected in its 3D structure. Therefore, we assume that C–S binding of BV to the GAF domain in miRFP670/C20A occurs via C3[1] and C3[2] as well. BV was solubilized in an aqueous buffer solution at pH 8.0 with dithiothreitol (DTT). It is known that BV has a ZZZsss conformation in solution and remains deprotonated at the pyrrolenine nitrogens in neutral/alkaline conditions[33]. In time-resolved experiments, an absorption or Raman spectrum of BV in solution was taken first, and apoprotein of miRFP670/C20A was added and stirred for 15 s. Afterwards transient absorption or Raman data were recorded every 60 s or ~15 s, respectively, until ~1 h (see "Methods" for details).

Figure 2a shows selected spectra of the time-resolved absorption experiments of miRFP670/C20A at pH 8.0. The gray line shows a BV spectrum in solution before the sample mixing, having two peaks around 390 nm (Soret band) and 670 nm (Q band). Immediately after the sample mixing (<1 min), a protein absorption peak appears at ~280 nm, and spectral evolution was observed in the visible-region (~400–700 nm) signals up to ~30 min, similarly to previously reported kinetics of other NIR FPs[23,31]. To quantify the spectral evolution, global analysis was applied. In global analysis, all wavelengths/wavenumbers are analyzed simultaneously with a set of common time contents following a sequential model $1 \rightarrow 2 \rightarrow 3 \rightarrow \dots$ that irreversibly interconvert with decreasing rate constants, which enables visualization of the spectral evolution. In Fig. 2b, a BV absorption spectrum before apoprotein addition and globally fitted transient absorption spectra (evolution-associated spectra; EAS, black, red, and blue lines) are shown. For the adequate fitting, three exponential components are required: 2.7 min, 19 min, and an infinite time component (>60 min). The indicated time constants denote the lifetime by which each EAS evolves into the next one. The first EAS (black line) is an EAS immediately after adding apoprotein of miRFP670/C20A (in 1 min). The first EAS (black) evolves to the second EAS (red) in 2.7 min. Subsequently, the second EAS (red) evolves to the third EAS (blue) in 19 min. The third EAS (blue) does not evolve any more within the time range of the experiments (~60 min). The gray line of Fig. 2b shows a steady-state BV absorption spectrum at pH 8.0 before adding apoprotein. In the transition from gray to black lines (<1 min), the Soret-band absorption intensity dropped by ~40% with a peak shift from 393 to 387 nm; while the Q-band absorption intensity increased by ~2.5-fold with a peak shift from 674 to 667 nm. In 2.7 min (transition from black to red, Fig. 2b), the Soret-band absorption shows only a ~3-nm blue shift and little spectral sharpening without intensity change; while the Q-band absorption increase by ~2-fold, peaking at ~671 nm. In 19 min (transition from red to blue, Fig. 2b), the Soret band absorption dropped only by 5% without a peak shift; while the Q-band

absorption increased by 30% with a significant peak shift to 643 nm. The final blue EAS did not evolve more under our temporal range (~60 min).

The time-resolved stimulated Raman spectra of miRFP670/C20A at pH 8.0 with a pre-resonance 800 nm Raman pump were globally fitted using the same time constants as the time-resolved absorption experiments (Fig. 2d–f). Supplementary Figure 1 shows the full Raman spectra of miRFP670/C20A in $H_2O$ and $D_2O$, with selected time traces. Notably, large H/D kinetics isotope effects (KIEs; ~1.3–2.0) were observed in the 2.7 and 19 min components. Figure 2c shows the steady-state stimulated Raman signal of BV at pH 8.0. The BV spectrum has strong peaks at $1255 \text{ cm}^{-1}$, which is assigned to a lactam mode[34,35], and at 1595, 1622, and $1650 \text{ cm}^{-1}$, which are assigned as C=C stretching modes[34,35]. Particularly, the C=C stretching modes are very sensitive to the isomeric form and protonation state[35], thus can be used for structural determination of BV in solution. Our BV Raman spectrum displays very high similarity to that of deprotonated ZZZsss conformer in solution[35]; therefore, we confirm that BV was in the ZZZsss form with deprotonated C-ring before adding apoproteins.

Immediately after adding the apoprotein of miRFP670/C20A to BV (<1 min component, evolution from gray to black lines, Fig. 2d–f), a substantial Raman signal increase was observed at 818, 1275, 1317, 1565, 1626, and $1647 \text{ cm}^{-1}$. Generally, the pre-resonance Raman intensity is dependent on the intensity and spectrum of an absorption band close to the Raman pump wavelength, in this case the Q-band absorption (at ~670 nm). As shown in Fig. 2b, the Q-band absorbance increased by 2.5-fold within <1 min (from gray to black lines). Thus, the ~2-fold Raman signal increase immediately after the sample mixing (from gray to black lines, Fig. 2d–f), such as at ~700 and $974 \text{ cm}^{-1}$, is considered as a result of an increase of the pre-resonance with the 800 nm Raman pump. However, the appearance of some particular Raman peaks such as at 818, 1275, and $1317 \text{ cm}^{-1}$ cannot be explained only by the increased resonance effect. Thus, these peaks result from BV conformational changes in <1 min.

In 2.7 min (evolution from black to red lines, Fig. 2d–f), strong peaks appeared at 679 and $713 \text{ cm}^{-1}$, in addition to ~2-3-fold peak increases at 818, 974, 1275, 1317, 1565, 1626, and $1649 \text{ cm}^{-1}$. Especially, the signals around $679 \text{ cm}^{-1}$ increased more than fourfold. Therefore, the $679 \text{ cm}^{-1}$ signal rise probably results from a structural change of BV in 2.7 min.

In 19 min (evolution from red to blue lines, Fig. 2d–f), a substantial Raman signal transition was observed in the C=C stretching region: the $1626\text{-cm}^{-1}$ peak is upshifted to $1630 \text{ cm}^{-1}$ with a ~20% signal loss, while the $1649\text{-cm}^{-1}$ band intensity rose by ~15% without a detectable peak shift. Overall, the Raman signal became weaker in 19 min, which probably resulted from a decreased resonance with the 800 nm Raman pump associated with the absorption blue-shift from 671 to 643 nm (Fig. 2b).

For detailed band assignments, quantum mechanical/molecular mechanics (QM/MM) calculations were performed based on the crystal structures of miRFP670[31] (see Methods for the detail) with a single C–S bond to Cys in the GAF domain (model I, Supplementary Figure 2) that simulates miRFP670/C20A, and with two C–S bonds to Cys residues in the PAS and GAF domains (model II, Supplementary Figure 2). Supplementary Figure 3 shows a comparison of experimental and calculated Raman spectra of miRFP670/C20A. Supplementary Table 1 shows detailed band assignments based on the QM/MM results.

**Isomerization and protonation within < 1 min.** Typically, the Soret and Q band absorption peaks and intensities are highly dependent on the conformation, the protonation state, and the

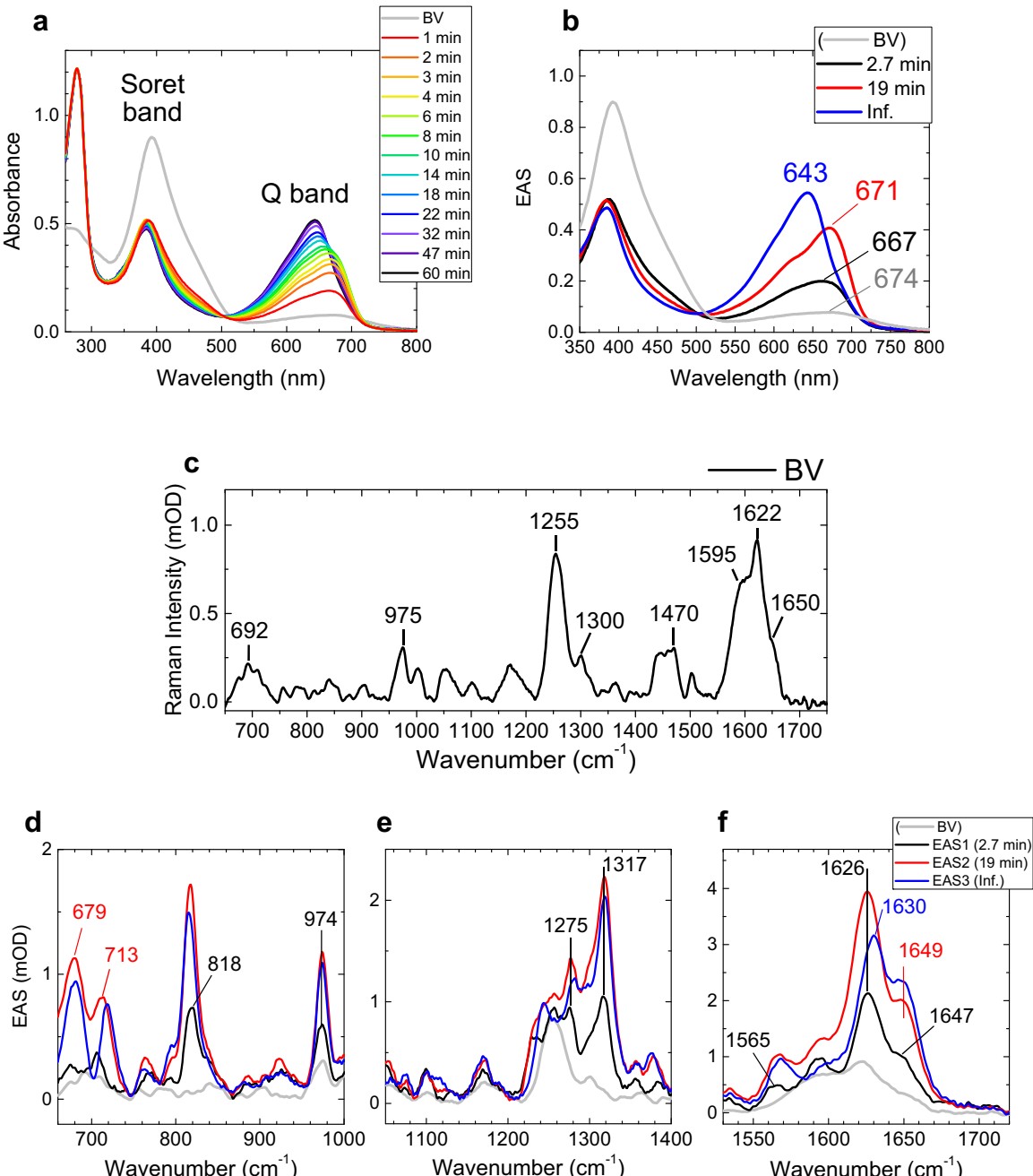

**Fig. 2 Time-resolved absorption and Raman spectra of miRFP670/C20A at pH 8.0. a** Selected time-resolved absorption spectra at the times indicated. **b** stimulated Raman spectrum of biliverdin (BV) prior to addition of apoprotein (gray line); Evolution-associated spectra (EAS) of globally fitted time-resolved absorption spectra. The indicated time constants denote the lifetime by which each EAS evolves into the next one. **c** Stimulated Raman spectrum of BV in a buffer solution with an 800 nm Raman pump. **d**–**f** EAS of globally fitted time-resolved stimulated Raman spectra with an 800 nm Raman pump. The gray lines denote the stimulated Raman spectrum BV in buffer solution reproduced from (**c**).

hydrogen-bond network of the chromophore[36–38]. Experimental and theoretical studies have indicated that linear tetrapyrroles in the ZZZsss form show a much stronger Soret-band absorption than Q-band absorption[36–38], which agrees with our interpretation that BV assumes a ZZZsss conformation in solution. As seen in Fig. 2b, the Soret-band absorption dropped by 40% in <1 min and did not change much afterwards. The substantial Soret-band absorption drop implies a BV conformation change from ZZZsss to another form, likely the final form ZZZssa, in <1 min. It has been reported that the Soret-band intensity, as compared with the Q-band intensity, decreases in the ZZZssa form[36–38], which is consistent with our observation. The C=C stretching region also

shows significant signal changes within 1 min (from gray to black, Fig. 2f); an upshift of the 1622 cm$^{-1}$ band to 1626 cm$^{-1}$, and a downshift of the 1650 cm$^{-1}$ band to 1647 cm$^{-1}$. According to our QM/MM calculation, the major contributions to these two bands are the C=C stretching of the C–D bridge and the ring D methyl group; hence these bands must be highly sensitive to the conformational change of the ring C and D. Therefore, the significant peak shifts of the ~1622 and ~1650 cm$^{-1}$ bands suggest conformational change around the D ring, supporting our interpretation that *sss–ssa* isomerization occurs within 1 min. Increase of the 818 cm$^{-1}$ bands, which is assigned to hydrogen-out-of-plane (HOOP) mode of the C- and D-rings (Supplementary

Table 1), implies that BV becomes highly distorted. Likely, BV isomerizes and localizes at the CBP already within <1 min, resulting in the large chromophore distortion as observed previously[23,31].

Protonation of BV is of significant importance influencing its NIR absorption and fluorescence properties[39,40]. In Raman spectroscopy, N–H in plane (ip) bending modes, which appear between 1550 and 1580 cm$^{-1}$, are used to assess the protonation state[41,42]. In the Raman spectrum of BV before adding the apoprotein, no peak was observed in the N–H ip region (~1550–1580 cm$^{-1}$, Fig. 2c), indicating that BV is deprotonated in solution at pH 8.0. Immediately after adding the apoprotein of miRFP670/C20A, a 1564-cm$^{-1}$ peak appears (black line, Fig. 2f). This 1564-cm$^{-1}$ peak is assigned to the N–H ip mode, implying protonation of BV occurs in <1 min. We also performed QM/MM calculation for the model I BV (Supplementary Fig. 2) with four protonated pyrrole nitrogens and with one deprotonated pyrrole nitrogen in ring B or C (Supplementary Fig. 4). The calculation shows that N–H ip mode appears at 1571 cm$^{-1}$ with the four protonated nitrogens (Supplementary Fig. 4a), while the N–H ip mode downshifts to <1540 cm$^{-1}$ with deprotonation of B or C ring (Supplementary Fig. 4b, c). The calculation supports our signal assignment of the 1564 cm$^{-1}$ peak to the N–H ip mode. Supplementary Fig. 5 shows normalized Raman spectra of the first and second EAS, demonstrating that the normalized relative N–H ip mode intensity hardly changed in the evolution in 2.7 min. Moreover, the N–H ip mode intensity remains almost unchanged in 19-min evolution (from red to blue evolution, Fig. 2f). These observations indicate that BV protonation is completed in <1 min after the sample mixing.

**Conformational change on B,C ring in 2.7 min**. In the 2.7-min evolution of the absorption spectra (from black to red EAS, Fig. 2b), a ratio of Q-band and Soret-band absorption (Q/Soret ratio) rose by ~2-fold with a 4 nm red shift of the Q-band absorption peak. Accordingly, the Raman intensity in the entire spectral region rose by ~2-fold in 2.7 min (from black to red EAS, Fig. 2d–f) because of the increase of resonance to the 800 nm Raman pump. In the normalized Raman spectra of the first and second EAS (Supplementary Fig. 5), the 1594 cm$^{-1}$ and ~1220–1280 cm$^{-1}$ signals dropped, whereas the ~680 cm$^{-1}$ signal increased in the 2.7-min evolution. The 1594 cm$^{-1}$ peak is mainly assigned to the C=C stretching mode on B–C methine bridge; the ~1220–1280-cm$^{-1}$ peaks include C–N stretching mode and N–H ip bending coordinates of the four pyrrole rings; and the ~680 cm$^{-1}$ band is attributed to N–H out-of-plane mode[43].

From mutagenesis and spectroscopic studies of Agp1[44] and *Dr*BphP[37] BphPs, it has been proposed that conserved His (His254 in miRFP670) and Asp (Asp201 of *Rp*BphP1, the parental protein of miRFP670, which is mutated to Thr201 in miRFP670) are the key residues for the hydrogen-bond network involving the BV pyrrole rings. Strikingly, His/Ala substitution (H250A and H260A for Agp1 and *Dr*BphP, respectively) decreased the Q/Soret ratio by ~2-fold with a few nm blue shift at a similar pH to our experimental condition (pH 7.8 for Agp1 and pH 8.0 for *Dr*BphP). Moreover, in resonance Raman spectra, a substantial increase of a ~1600 cm$^{-1}$ peak was observed in both of Agp1 and *Dr*BphP in the His/Ala mutants[37,44]. From these observations, we suggest that the Q/Soret ratio and the 1600 cm$^{-1}$ peak are affected by the hydrogen-bond network near the pyrrole rings. The first and second EAS of our time-resolved absorption and Raman spectra show very similar spectral differences to His/Ala mutants and wild-type of Agp1 and *Dr*BphP: ~2-fold Q/Soret ratio increase, a 4 nm red-shift of the Q-band absorption peak (Fig. 2b), and decrease of the ~1594-cm$^{-1}$ Raman peak (Supplementary Fig. 5).

These similarities suggest that rearrangement of hydrogen-bond network proceeds in 2.7 min on BV binding to miRFP670/C20A. The large signal changes at ~680 cm$^{-1}$ (N–H out-of-plane) and ~1220–1280-cm$^{-1}$ (C–N stretching mode and N–H ip bending coordinates) signals support a hydrogen-bond rearrangement in 2.7 min. The observation of a significant H/D exchange effect on the 2.7 min time constant (Supplementary Fig. 1) is consistent with this assignment.

We also note that a C–S covalent bond was not formed after 2.7 min yet. If a C–S covalent bond would be formed between BV and Cys in the GAF domain within 2.7 min, a spectral blue-shift to ~640 nm must be detected, which contradicts our observation.

**Covalent thioether bond formation in 19 min**. We showed that the BV isomerization and protonation occurs in <1 min, hydrogen-bond rearrangement proceeds in 2.7 min but no C–S bond is formed on these time scales. In the time-resolved absorption data, a blue-shift of the Q-band absorption peak from 671 to 643 nm in 19 min (transition from red to blue line, Fig. 2b) was observed. In BphP1-FP, which is the first NIR FP engineered from *Rp*BphP1, C–S bond formation to Cys253 in the GAF domain causes a ~30 nm blue shift[23]. This observation implies that C–S bond formation to Cys253 in the GAF domain, via C3$^1$ or C3$^2$, proceeds in 19 min in miRFP670/C20A after mixing BV and apoprotein. In the time-resolved Raman spectra, a signal drop at 1620 cm$^{-1}$ was detected with a signal rise at ~1650 cm$^{-1}$ in 19 min, resulting in a larger shoulder peak at 1649 cm$^{-1}$ (Figs. 2f and S6). Based on our QM/MM simulations, we assign those bands to C=C stretching of a methine bridge between the A–B rings (1620 cm$^{-1}$); ring-D ethyl and a minor contribution from a methine bridge between the C–D rings (1630 cm$^{-1}$); and C–D methine bridge and a minor contribution from ring-D ethyl (1649 cm$^{-1}$) (Supplementary Table 1). Notably, in double Cys mutants of iRFP682, which has no C–S covalent bond of BV, a clear peak at 1619 cm$^{-1}$ was detected, while the 1619 cm$^{-1}$ signal was not seen when BV is covalently bound to Cys in the GAF domain[32]. These observations indicate that the 1619 cm$^{-1}$ and ~1650 cm$^{-1}$ bands can be used as an indicator of C–S bond formation to the GAF domain. Hence, the 1620 cm$^{-1}$ signal drop (Figs. 2f and S6) supports the C–S bond formation to the GAF domain in 19 min. The observation of a significant H/D exchange effect on this time constant (Supplementary Fig. 1) suggests that proton or hydrogen transfers are rate limiting in the C-S bond formation reaction. In Fig. 3, the overall reaction model of BV to miRFP670/C20A binding is shown.

**Time-resolved experiments on miRPF670**. Next, we studied BV incorporation dynamics in miRFP670. Figures 4a, b show selected time-resolved absorption spectra and globally fitted EAS of absorption with four time components; 1.5 min, 4.3 min, 14 min, and infinite (>60 min). The Soret-band absorption peak around ~400 nm dropped by ~40% in <1 min similar to the miRFP670/ C20A mutant, whereas a Q-band absorption peak appeared at ~663 nm (black line, Fig. 4b). In 1.5 min, the Q-band absorption increased with a ~10-nm red-shift; a similar red-shift was also observed in the C20A mutant in 2.7 min (Fig. 2b). Moreover, a large blue-shift from 673 to 647 nm was observed in 4.3 min in miRFP670 (from red to blue, Fig. 4b) that is comparable to the 19-min transition in the C20A mutant (Fig. 2b). The blue-shifted Q band absorption increased in 14 min in miRFP670 with only a small peak shift (Fig. 4b), which was not detected in the time-resolved absorption in the miRFP670/C20A mutant (Fig. 2b).

Figures 4c–e and S7 show EAS of time-resolved stimulated Raman data of miRFP670, globally fitted using the same time constants with the EAS of time-resolved absorption. Peak

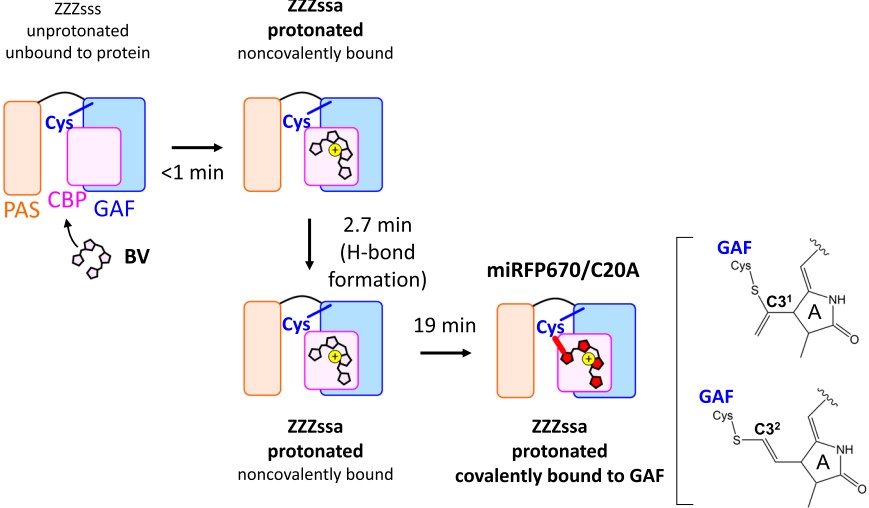

**Fig. 3 Biliverdin (BV) incorporation model of miRFP670/C20A.** Within 1 min after adding the miRFP670/C20A apoprotein to the biliverdin (BV) solution, BV localization to the chromophore binding pocket (CBP) occurs, binding noncovalently to the protein. BV isomerization and protonation also proceeds within 1 min. In 2.7 min, a hydrogen-bond rearrangement of BV occurs. In 19 min, C–S covalent bond formation between BV and Cys253 in the GAF domain via $C3^1$ or $C3^2$ proceeds, completing the BV binding reaction to miRFP670/C20A.

positions and signal developments of the first three EAS in miRFP670 are similar to those of the miRFP670/C20A mutant, while some peak intensities are different (Fig. 2d–f). Supplementary Fig. 8 shows time traces of miRFP670 and its C20A mutant at selected wavenumbers. Additional spectral evolution is detected in 14 min in miRFP670 (from blue to green, Fig. 4c–e), which was absent in the miRFP670/C20A mutant. Difference spectra of third and second EAS, and fourth and third EAS are shown in Supplementary Fig. 9. In the difference spectrum of the fourth and third EAS, two strong peaks at ~1630 and ~1650 cm$^{-1}$ are observed. Remarkably, the intensity of the ~1650 cm$^{-1}$ signal is nearly 90% of the ~1630 cm$^{-1}$ peak, which is much higher than other EAS in Fig. 4e.

**Isomerization, protonation, and thioether bond formation to Cys in GAF.** The similarity of the first, second and third EAS of absorption and Raman in miRFP670 (Fig. 4) and its C20A mutant (Fig. 2) implies that the first three reactions of miRFP670 proceeding in <1, 1.5, and 4.3 min can be assigned to the similar reactions occurring in <1, 2.7, and 19 min in the miRFP670/C20A mutant (Fig. 3), i.e., isomerization, localization to the CBP and pyrrolenine nitrogen protonation in <1 min, hydrogen-bond rearrangement in 1.5 min, and a C–S bond formation to the GAF domain in 4.3 min. One may consider that the 4.3 min reaction could be involved with a thioether bond formation to the PAS domain; however, it was demonstrated that mutants of NIR FPs having a single C–S bond to the PAS domain showed red-shifted absorption similar to noncovalently bound BV[23,32,45]. This contradicts our observation that the Q-band absorption was blue-shifted in 4.3 min; hence we exclude the possibility that a thioether bond to the PAS domain was formed in this time scale. The substantial kinetic difference between miRFP670 and its C20A mutant in the first three EAS suggests that Cys in the PAS domain influences the structure near CBP and facilitates thioether bond formation.

**Formation of the 2nd C–S bond to the PAS domains in 14 min.** As observed in Fig. 4e and S8, C=C stretching Raman signals at ~1630 and ~1650 cm$^{-1}$ increased by ~30% in 14 min. In the Q-band absorption, a ~30% signal rise was detected in miRFP670 in 14 min (Fig. 4b); hence the Raman signal increase in 14 min is explained by Q-band absorption increase resulting in higher resonance to the 800 nm Raman pump. Significantly, as seen in Supplementary Fig. 9, the shoulder at ~1650 cm$^{-1}$ is much higher than in the third EAS (blue line, Fig. 4e). In purified iRFP682, in which most BV have two C–S bonds, it was shown that the ratio of the ~1650 cm$^{-1}$ signal to the ~1630 cm$^{-1}$ signal is ~1.3-fold higher than the Cys mutant in the PAS domain[32]. Therefore, the Raman data of iRFP682 implies that when BV binds to both of the PAS and GAF domains simultaneously, the upshifted ~1650 cm$^{-1}$ band intensity is enhanced. This feature corresponds to our observation in the Raman spectra (Fig. 4e and S9), implying that the formation of the 2nd C–S bond to the PAS domain occurs in 14 min. Likely, the extinction coefficient increases with dual C–S bond formation in miRFP670 as observed in iRFP682[32], which results in the ~30% increase of the Q-band absorption. In Fig. 4f, the BV binding model of miRFP670 is summarized.

**Formation of the protein with dual Cys bonds.** To further clarify our model for BV binding in miRFP670, we analyzed the protein on SDS-PAGE at different time points during the chromophore assembly reaction. As was shown previously[31], protein species with BV bound to both Cys in the PAS and in the GAF appear as a separate faster running bands on SDS-PAGE. This happens because there is a topologically closed loop in the protein with double-bound BV that cannot be linearized by denaturation. We found that double-bound protein species were not formed during first 3 min of assembly reaction, which is consistent with the recorded spectra that do not show blue-shifted absorption at this time frame (Supplementary Fig. 10a, e). When we further studied the timing of double-bound protein formation, we detected its parallel formation together with the protein species with the BV chromophore bound to Cys in the GAF domain (Supplementary Fig. 10b, f).

**Time-resolved experiments on miRFP709.** To assess the binding dynamics in miRFPs in the presence of a Cys in the PAS domain only, and not in the GAF domain, we further studied BV incorporation dynamics in miRFP709. miRFP709 is derived from the same template as miRFP670, RpBphP1, and apart from the presence/absence of a Cys in the GAF domain, it differs from miRFP670 by only two amino acids.[31] The position of the Cys in

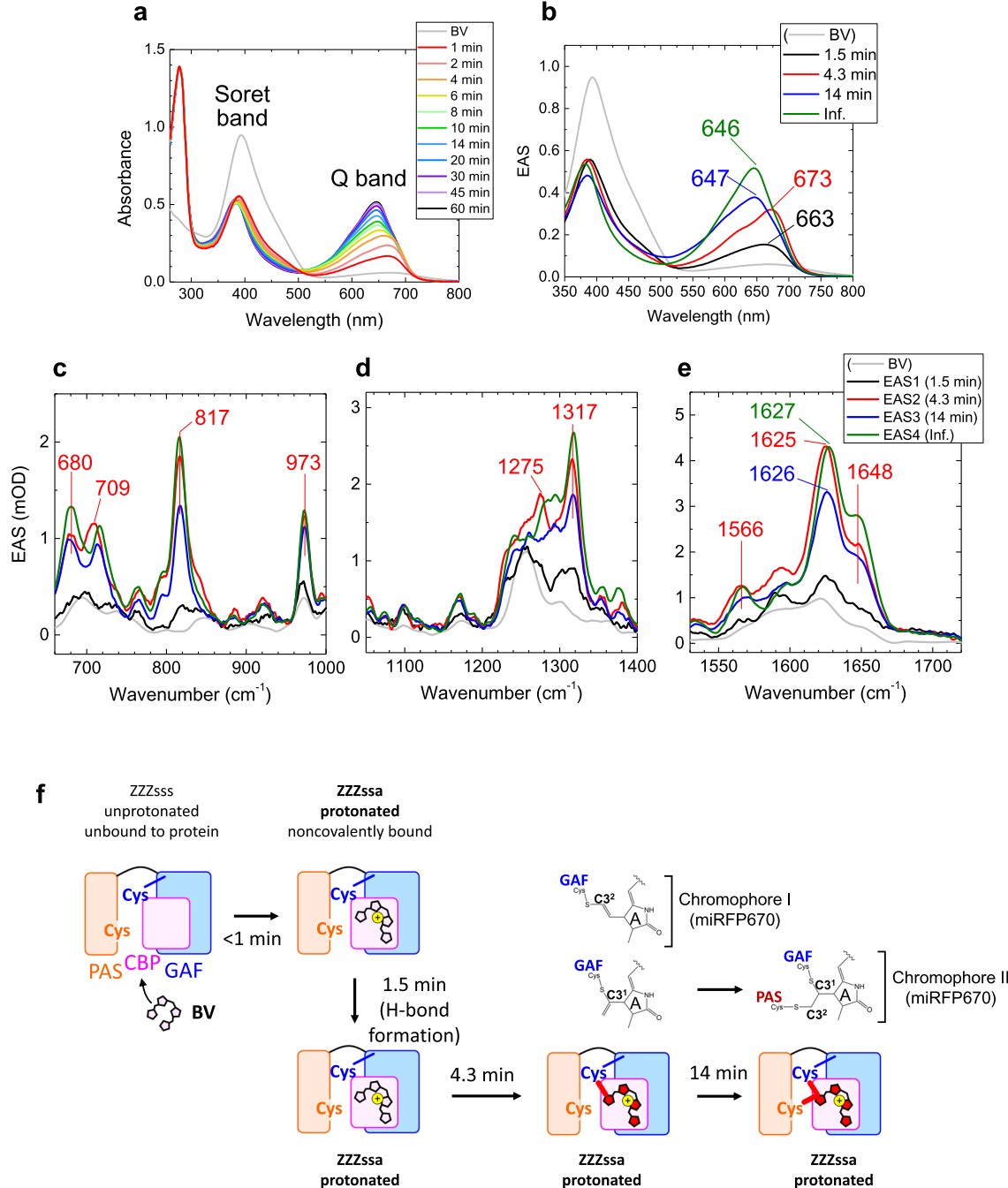

**Fig. 4 Time-resolved absorption and Raman spectra of miRFP670 at pH 8.0 and incorporation model of miRFP670. a** Selected time-resolved absorption spectra taken at the times indicated. **b** Evolution-associated spectra (EAS) of globally fitted time-resolved absorption spectra. **c–e** EAS of globally fitted time-resolved stimulated Raman spectra with an 800 nm Raman pump. The indicated time constants denote the lifetime by which each EAS evolves into the next one. **f** Biliverdin (BV) incorporation model of miRFP670.

the PAS domain is identical to that of miRFP670 and native BphPs. Consequently, its absorption and emission spectra peak at 683 and 709 nm, respectively, similar to that of native BphPs and significantly red-shifted with respect to that of miRFP670. Figure 5a shows selected time-resolved absorption spectra up to 60 min after adding apoprotein of miRFP709 to BV solution. The time-resolved absorption spectra were globally fitted with three-time components: 2.1 min, 18 min, and an infinite component (>60 min.) (Fig. 5b). The Soret-band absorption intensity dropped by ~40% immediately after adding the apoprotein (in <1 min). Also, the Q-band absorption rose up in <1 min, peaking

at 674 nm (black line, Fig. 5b). In 2.1 min, the Q-band absorption rose up by ~30% with a 2-nm peak shift to 676 nm (from black to red, Fig. 5b). In 18 min, the Q-band absorption rose up more (by ~40%) with a peak shift to 683 nm (from red to blue, Fig. 5b), which did not evolve any more in the experimental temporal range.

Figure 5c–e and S11 show globally fitted time-resolved stimulated Raman spectra of miRFP709 after adding the apoprotein to BV solution, fitted with the same time constants as the time-resolved absorption spectra in Fig. 5b. Immediately after the apoprotein mixing, a peak at 1561 cm$^{-1}$ increased,

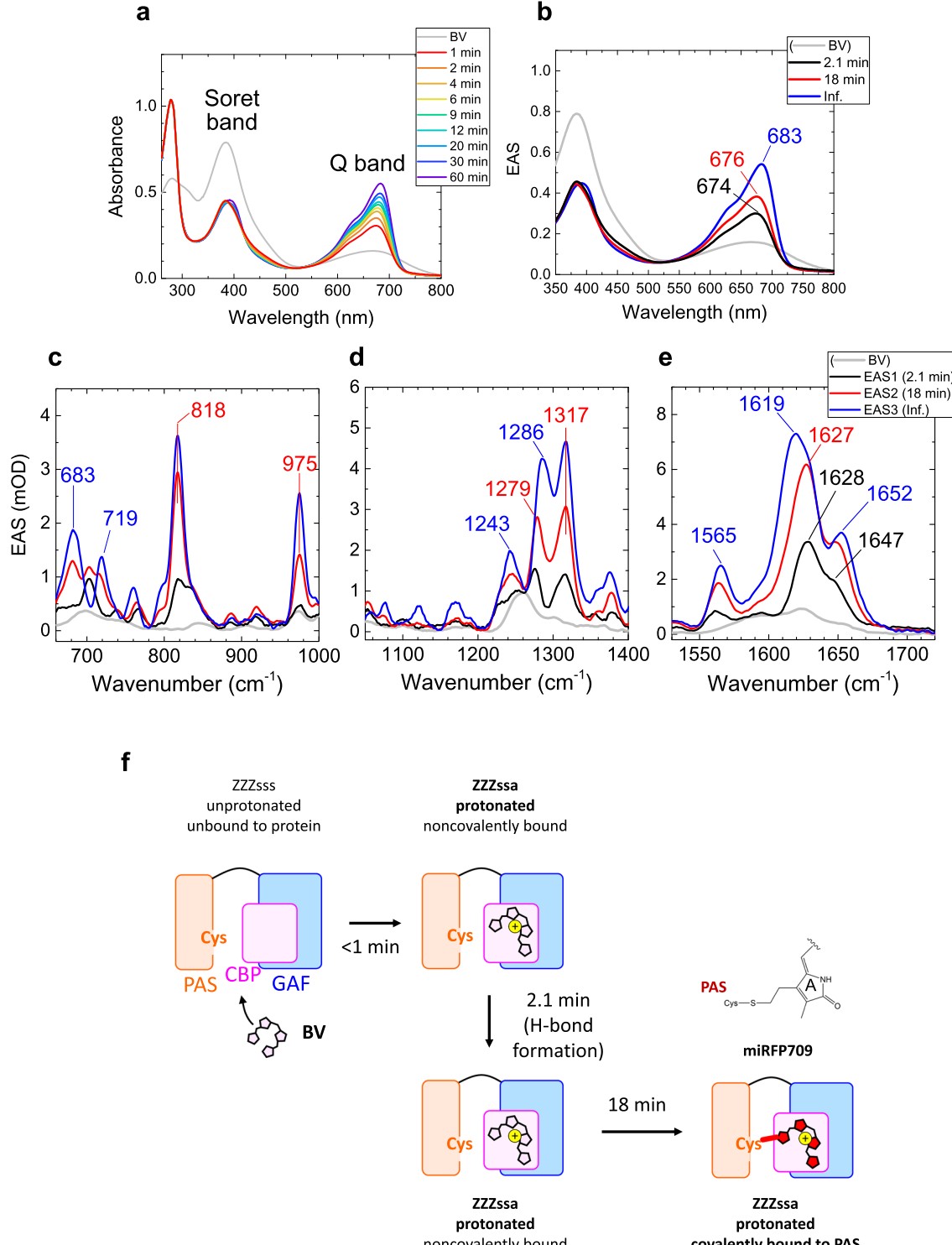

**Fig. 5 Time-resolved absorption and Raman spectra of miRFP709 at pH 8.0 and biliverdin (BV) incorporation model of miRFP709. a** Selected time-resolved absorption spectra. **b** Evolution-associated spectra (EAS) of globally fitted time-resolved absorption spectra. **c–e** EAS of globally fitted time-resolved stimulated Raman spectra with an 800 nm Raman pump. The indicated time constants denote the lifetime by which each EAS evolves into the next one. **f** BV incorporation model of miRFP709.

similar to miRFP670 (Fig. 4e) and its C20A mutant (Fig. 2f), which is assigned to an N–H in plane (ip) bending mode, a marker of BV protonation (from gray to black, Fig. 5e). We assign this fast reaction within 1 min to BV isomerization, protonation, and localization to the CBP of miRFP709, as we suggest in miRFP670 and its C20A mutant. In 2.1 min, the Raman signals

rose in the entire spectral region (from black to red, Fig. 5c–e), resulting in a similar Raman spectrum to that of miRFP670 and its C20A mutant (2nd EAS, red lines in Figs. 2f, 4e, and 5e). The Raman spectral similarity of the 2nd EAS and the similar time constant (1.5 and 2.7 min for miRFP670 and its C20A mutant, respectively) imply that the reaction in 2.1 min is hydrogen bond

formation/rearrangement, as in miRFP670 and its C20A mutant. In 18 min, the Raman signal intensities further increased (from red to blue, Fig. 5c–e), consistent with the increase of the absorption. Also Raman spectral shifts were detected at ~1600–1650 cm$^{-1}$ in 18 min, indicating that the C=C backbone structure changed. We assign the 18-min reaction to the covalent bond formation to Cys in the PAS domain. We note that such relatively slow binding dynamics to Cys in PAS (18 min) are consistent with the slow process in miRFP670 (14 min), which was assigned to Cys in PAS binding as well. We conclude that covalent binding to Cys in PAS is an intrinsically slow process. In Fig. 5f, the BV binding model of miRFP709 is shown.

## Discussion

Using the sub-minute time-resolved absorption and preresonance stimulated Raman spectroscopies we have determined the BV chromophore binding dynamics in NIR FPs, such as miRFP670, its C20A mutant and miRFP709 (Figs. 3, 4f and 5f). We have found that in <1 min, the BV chromophore undergoes isomerization with localization to the CBP of the protein and pyrrolenine nitrogen protonation. Next, the hydrogen-bond rearrangement of the BV chromophore proceeds in 1.5, 2.7, or 2.1 min in miRFP670, its C20A mutant, and miRFP709, respectively. Following the C–S bond formation to the Cys in the GAF domain occurs in 4.3 min (miRFP670) and 19 min (miRFP670/C20A), while a C–S bond is formed to the Cys in the PAS domain in 18 min in miRFP709. In miRFP670, there are two competing processes: one is BV binding to Cys in the GAF via C3$^2$, which appears to be a stationary product, the other one is BV binding to two Cys in the PAS via C3$^2$ and in the GAF via C3$^1$.

The fluorescence quantum yield and extinction coefficient are higher in miRFP670 as compared to the C20A mutant[31]. The favorable properties of miRFP670 were assigned to the dual C-S bond formation, which increases the rigidity of the BV chromophore. This idea is consistent with our previous proposal that a diminished ring A mobility could increase the fluorescence quantum yield[45].

An important result from the present study is the direct evidence that the presence of the Cys in the PAS domain facilitates the formation of the C-S bond to the Cys in GAF, as this process occurs four times faster in miRFP670 as compared to the miRFP670/C20A mutant. Time-resolved experiments allowed us to observe two parallel processes happening during BV binding to the miRFP670 apoprotein, as shown in Fig. 4f. The BV binding to Cys in the GAF via C3$^2$ appears to be a stationary product, as these protein species cannot sterically bind to the Cys in the PAS located outside of the chromophore pocket, as the distal binding carbon C3$^2$ is already occupied (Fig. 1).

Overall, these findings open up the real-time structural analysis of chromophore binding in BphPs and BphP-derived NIR FPs, and may provide insights for the rational design of blue-shifted NIR FPs. The presence of two chromophore binding Cys appears to not only increase the quantum yield of the FP, as was shown previously[32], but also to speed the reaction of chromophore binding to the apoprotein. Faster BV binding should result in better competition with non-productive binding with other tetrapyrroles, such as protoporphyrin IX, resulting in higher concentration of BV-bound protein in a cell and observed superior cellular brightness of double-Cys BphP-derived NIR FPs.

## Methods

**Protein expression and purification.** The miRFP670 and miRFP709 genes were cloned into a pBAD/His-B vector (Invitrogen). Site-specific mutagenesis of miRFP670 was performed using QuikChange kit (Stratagene), resulting in miRFP670/C20A mutant. The miRFP670, miRFP670/C20A and miRFP709 proteins with polyhistidine tags on the N-termini were expressed in LMG194 bacterial cells (Invitrogen) bearing the pWA23h plasmid encoding a heme oxygenase under the rhamnose promoter[5]. To initiate protein expression, the bacterial cells were grown in RM medium supplemented with ampicillin, kanamycin and 0.02% rhamnose for 5 h at 37 °C. Then 0.002% arabinose was added, and the bacterial culture was incubated for an additional 12 h at 37 °C followed by 24 h at 18 °C. The resulting holoproteins were purified using a Ni-NTA agarose (Qiagen). An Ni-NTA elution buffer contained 100 mM EDTA and no imidazole. After elution, the buffer was substituted with phosphate-buffered saline using PD-10 desalting columns (GE Healthcare). In experiment with apoproteins, they were purified without co-expression of the heme oxygenase and then dissolved in 20 mM Tris-HCl buffer with 150 mM NaCl at pH or pD 8.0. The protein concentrations were ~0.45 and ~0.9 mM for time-resolved absorption and Raman experiments, respectively.

**Preparation of the biliverdin (BV) solution.** A BV stock solution was prepared in 20 mM Tris-HCl buffer with 150 mM NaCl at pH or pD 8.0 with the concentration of 0.1 mM (with 2 mM DTT) and 0.5 mM (with 10 mM DTT) for time-resolved absorption and Raman experiments, respectively.

**Time-resolved absorption spectroscopy.** The time-resolved absorption spectroscopy measurements were performed for miRFP670, miRFP670/C20A, and miRFP709 at pH 8.0 by using a UV–Vis spectrometer (Cary 4000, Agilent). The detection wavelength range was set at 260–800 nm. 40 µL the BV solution was filled in a 3-mm pathlength quartz cuvette (105.251-QS, Hellma Analytics). A UV–Vis absorption spectrum of BV was first measured. Subsequently, 40 µL of the apoprotein sample solution was added and stirred by a pipette for 15 s. Immediately after the sample mixing, the UV–Vis absorption spectra were obtained every 1 min up to 60 min.

**Time-resolved watermarked stimulated Raman spectroscopy.** The time-resolved stimulated Raman spectroscopy measurements of miRFP670, miRFP670/C20A, and miRFP709 were performed at pH or pD 8.0 by using a previously reported watermarked stimulated Raman spectroscopy setup[16–19]. The Raman pump (800 nm, ~3 µJ) and supercontinuum Raman probe (~840–960 nm) were spatiotemporally overlapped at the sample position. The diameter of the Raman pump and probe was ~100 µm (FWHM). Raman signals at 650–1800 cm$^{-1}$ were analyzed. Forty microliters of a BV solution was filled in a 3-mm pathlength quartz cuvette (105.251-QS, Hellma Analytics). A stimulated Raman spectrum of BV was first measured. Subsequently, 40 µL of the apoprotein sample solution was added and stirred by a pipette for 8 s. Immediately after the sample mixing, stimulated Raman spectra were obtained every 15.2 s up to ~50 min. For each time point, the Raman signal was accumulated for 8 s. During the experiments, a home-built vibrational stage was used to move the sample. For the obtained Raman spectra, baseline correction by the watermarking approach[16] was applied, and the remaining minor baselines were corrected manually.

**Global analysis methodology.** Global analysis was performed for the time-resolved absorption spectra and stimulated Raman spectra using the Glotaran program[46]. With global analysis, all wavelengths/wavenumbers were analyzed simultaneously with a set of common time constants[47]. A kinetic model was applied consisting of sequentially interconverting, EAS, i.e., $1 \rightarrow 2 \rightarrow 3 \rightarrow \ldots$ in which the arrows indicate successive mono-exponential decays of a time constant, which can be regarded as the lifetime of each EAS[47]. The first EAS corresponds to the difference spectrum at time zero. The first EAS evolves into the second EAS with time constant $\tau_1$, which in turn evolves into the third EAS with time constant $\tau_2$, etc. The procedure clearly visualizes the evolution of the intermediate states of the protein[48]. Note that a global analysis in terms of sequentially interconverting species is mathematically equivalent to a sum-of-exponentials global analysis[49]. The estimated standard errors in the time constants were 10%.

**Theoretical calculations.** The structure of the chromophore binding site miRFP670 proteins was optimized at a hybrid quantum chemical/molecular mechanic (QM/MM) level using the as starting geometry the corresponding crystallographic coordinates (PDB entry: 5VIV)[31]. Two models were generated considering the two chromophore species detected in the electron density maps: model I, characterized by a single covalent bond between BV- C3$^2$ and the Cys253 of the GAF domain and model II, characterized by two covalent bonds between BV-C3$^1$ and Cys253 and between C3$^2$ and Cys20 from the PAS domain (Supplementary Fig. 2). Hydrogens were added via the HBUILD package implemented in the CHARMM V39.2 software operating on GPUs[50]. His254 and His284 in the CBP were modeled as charge neutral residues with a hydrogen at the Nε site. The BV moiety in models I and II were protonated at all four pyrrole nitrogens, yielding a total charge of −1e. In the case of model I, two additional protonation states of the BV molecule were considered: one with a deprotonated nitrogen at ring B and one with a deprotonated nitrogen at ring C (total charge of −2e). The entire protein was then solvated in a 30 × 30 × 30 Å$^3$ box of TIP3P water containing 150 mM chloride and sodium ions. Keeping the coordinates of the protein fixed, the conformation of the water molecules was optimized by energy minimization followed by short, 50 ps long, classical molecular dynamic simulation using the CHARMMv39.2 software[50] and the CHARMM36 force field for the protein[51].

Geometry optimization of the CBPs were performed on reduced systems which were constructed by extracting from the last snapshot of the MD simulation spheres containing all residues within a radius of 40 Å from nitrogen atom at pyrrole ring C. All atoms confined within a concentric sphere of 20 Å were defined as 'active atoms' and allowed to move upon energy minimization following either the laws of quantum mechanics or molecular mechanics. The remaining atoms where held fixed. In the active region, the BV chromophore, the pyrrole water, Iso202 and the side chains of His254, His284, Tyr170, Tyr197, Phe257, and Cys253 were treated at the B3LYP/6-31 G* level of theory (182 atoms in total) while the interactions involving the remaining molecules were described with the CHARMM36 force field. In the case of model II, the side chain of Cys20 was also treated quantum mechanically. The coupling between the QM and MM regions was modeled using the electrostatic embedding model[52] combined with the charge shifted scheme and the covalent cuts at the QM/MM boundary were saturated with hydrogens. The QM/MM geometry optimization of the active region was performed with a limited memory L-BFGS quasi Newton optimization algorithm[53] working with hybrid delocalized internal coordinates implemented in the modular program package Chemshell[54]. The optimized structures of model I and model II were used as input for the calculation of Raman spectra using the computational approach described in an earlier publication[55].

**SDS-PAGE analysis during BV assembly reaction**. For kinetic analysis of BV assembly with NIR FP apoproteins, purified apoprotein (15 mM) was mixed with 10 mM BV in PBS containing 1 mM DTT. Absorbance spectra were monitored with a Hitachi U-2000 spectrophotometer immediately after mixing (time 0 min) and at the indicated time points. For protein gel, 10 μg of the protein were taken from the kinetic reaction at indicated time points and added to the SDS-PAGE loading buffer to stop the reaction, then samples were immediately placed at 95 °C for 5 min and loaded on the 4–20% Mini-PROTEAN TGX gel, BioRad. After run, the gel was stained with the GelCode Blue Safe Protein Stain, ThermoFisher Scientific.

## Data availability

The main data supporting the findings of this study have been deposited in repository DataverseNL with identifier https://doi.org/10.34894/5LLEWN. The additional data are available from the corresponding author on reasonable request.

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

## Acknowledgements

This work was supported by VICI grant from the Chemical Sciences Council of the Netherlands Organization for Scientific Research (NWO) and Middelgroot investment grant (both to J.T.M.K.), GM122567 and NS103573 grants (both to V.V.V.), and EY030705 grant (to D.M.S.) from the US National Institutes of Health, and 322226 from the Academy of Finland. M.A.M. thanks the German Research Foundation (DFG – SFB1078 project C2) for funding.

## Author contributions

M.B. and D.M.S. developed and purified the miRFPs proteins; Y.H., J.W., and M.K. established the stimulated Raman setup; Y.H. and S.A.J. carried out the time-resolved absorption and Raman experiments; Y.H. performed the global analysis for the time-resolved data; F.V.E. and M.A.M performed and analyzed the QM/MM calculation; V.V.V. and J.T.M.K. designed the project; Y.H and J.T.M.K. wrote the manuscript with contribution from all co-authors. All authors reviewed the manuscript.

## Competing interests

The authors declare no competing interests.
