## [Peer Review File · Communications Chemistry]

Reviewers' comments:

Reviewer #1 (Remarks to the Author):

In this manuscript, Hontani et al. studied the binding of the tetrapyrrole biliverdin chromophore to the engineered phytochrome, which are near infrared fluorescent proteins. They used time-resolved stimulated Raman technique to monitor the structural dynamics of the biliverdin while it binds to the protein. Then, they concluded how the chromophore reacts in the binding pocket from their novel observations. Their experimental data, data analysis, and interpretation are mostly good and reasonable.

However, the weak point of this study is that their observations are not directly related to the protein function, which is the fluorescent property, or its utility. So, I think that their study is good, but does not provide a sufficient impact.

I also have some comments as below.

Line 70-77:

In the introduction, the authors compare stimulated Raman and conventional Raman methods, describing that conventional Raman spectroscopy usually requires cryogenic temperature and hours for signal averaging to measure high-quality ground-state spectra. However, I think that, using current instruments, we can measure good Raman spectra within a minute by conventional Raman method. Rather, I think that the advantage of stimulated Raman method is the reduction of fluorescence background especially to measure pre-resonance ground-state Raman spectra, because stimulated Raman signal has the direction.

Line 197-200:

The authors described that NH bending mode at 1550-1580 cm^{-1} are used to assess the protonation state. They should cite the reference. A recent study reported that the tetrapyrrole chromophore deprotonated at B ring also exhibit the NH bending mode at 1541 cm^{-1} (S. Osoegawa et al., J. Phys. Chem. B, 2019, 123,3242-3247.)

Figure 3:

The model includes two kinds of the bound biliverdin.

I wondered how these two are deduced from their experiment.

Figure 4:

Peak positions should be written in c,d.

Line 433

H254 and H284 → His254 and His284 (for consistency in the main text)

Reviewer #2 (Remarks to the Author):

This is an interesting paper that investigates the binding of the BV chromophore to the apoprotein of a near-infrared fluorescent protein (miRFP670) developed from a bacteriophytochrome. In particular, using time-resolved stimulated Raman spectroscopy techniques, the paper presents real-time data that offer new insights into this process, which are likely to be useful in current efforts to increase the fluorescence quantum yields of such proteins and thereby facilitate their application as bioimaging tools. Specifically, the paper provides valuable details of how the binding of the BV chromophore to two cysteine residues (one in the PAS domain and one in the GAF domain) of miRFP670 occurs. Given that this dual binding (through two thioether bonds) is a key feature separating miRFP670 from very weakly fluorescent native bacteriophytochromes, which

bind the BV chromophore through a single cysteine residue, it has been argued that the increased rigidity offered by the additional thioether bond plays an important role in increasing the fluorescence quantum yield of miRFP670. Accordingly, it is worthwhile to gain a better understanding of the binding of the BV chromophore to the two cysteine residues, as provided by the present paper.

Still, I think there are two issues that need to be addressed in a revised manuscript before this paper can be considered for publication in Communications Chemistry.

(1) First, while the paper considers the native miRFP670 protein and a mutant lacking the cysteine residue in the PAS domain, it seems to me that a better appreciation of the problem at hand would also require comparison with a mutant lacking the cysteine residue in the GAF domain.

(2) I think parts of the presentation contain perhaps too much "spectroscopic jargon" and are a little bit too vague to be fully appreciated by a broad readership. This holds true especially for the Section "Time-resolved experiments in the miRFP670/C20A mutant". For example, on page 6, these points are illustrated by the sentences:

"As shown in Fig. 2b, the Q-band intensity was increased by 2.5-fold. Thus, the ~ 2 -fold signal increases such as at ~ 700 and 974 cm^{-1} are considered as a result of an increase of the pre-resonance with the 800 nm Raman pump. However, the appearance of some peaks such as at 818 , 1275 and 1317 cm^{-1} cannot be explained only by the higher resonance effect; thus, these peaks are derived from BV conformational change in $<1\text{ min.}$ "

The level of detail provided is sufficient from the point of view of reproducibility.

We thank the reviewers for their critical reviews and insightful remarks: their comments were very helpful to improve the manuscript. In the manuscript, the amended parts are highlighted in green, as well as the responses to the reviewers' comments below.

Reviewers' comments:

Reviewer #1 (Remarks to the Author):

In this manuscript, Hontani et al. studied the binding of the tetrapyrrole biliverdin chromophore to the engineered phytochrome, which are near infrared fluorescent proteins. They used time-resolved stimulated Raman technique to monitor the structural dynamics of the biliverdin while it binds to the protein. Then, they concluded how the chromophore reacts in the binding pocket from their novel observations. Their experimental data, data analysis, and interpretation are mostly good and reasonable.

We thank the reviewer for the favorable assessment of our work.

However, the weak point of this study is that their observations are not directly related to the protein function, which is the fluorescent property, or its utility. So, I think that their study is good, but does not provide a sufficient impact.

It is the structure of the complex, the nature of the covalent binding and the type of chromophore in this pigment-protein complex that determines the fluorescence properties. We emphasize that studying these proteins with fluorescence spectroscopy (which the reviewer seems to suggest) would not give any information regarding the molecular factors that determine its fluorescence function. Here, we assess structural aspects and their dynamics of chromophore binding to the apoprotein, and hence our results are directly relevant to its fluorescence function. For instance, we found that the presence of the Cys in the PAS domain facilitates formation of the C-S bond to the Cys in the GAF domain. The faster BV binding contributes to a higher BV binding efficiency when compared to binding of less-fluorescent tetrapyrroles. In addition, we find that formation of the second covalent bond to the PAS domain is the final step in the binding process, which leads to improved fluorescence properties. In this regard, our findings are directly related to the protein's fluorescence function.

I also have some comments as below.

Line 70-77:

In the introduction, the authors compare stimulated Raman and conventional Raman methods, describing that conventional Raman spectroscopy usually requires cryogenic temperature and hours for signal averaging to measure high-quality ground-state spectra. However, I think that, using current instruments, we can measure good Raman spectra within a minute by conventional Raman method. Rather, I think that the advantage of stimulated Raman method is the reduction of fluorescence background especially to measure pre-resonance ground-state Raman spectra, because stimulated

Raman signal has the direction.

We followed the advice of the reviewer and deleted the remark on hourslong data collection times at cryogenic temperatures, and emphasized the insensitivity of the stimulated Raman method to fluorescence on line 74, p. 3.

Line 197-200:

The authors described that NH bending mode at 1550-1580 cm^{-1} are used to assess the protonation state. They should cite the reference. A recent study reported that the tetrapyrrole chromophore deprotonated at B ring also exhibit the NH bending mode at 1541 cm^{-1} (S. Osoegawa et al., J. Phys. Chem. B, 2019, 123,3242-3247.)

We performed QM/MM calculation for BV in the model I (miRFP670/C20A) with the four pyrrole N protonated and BV with a deprotonated N at ring B or C (Fig. S4, below). When the four rings are protonated, the N–H ip band appears at 1571 cm^{-1} . When one of the two inner (B or C) rings is protonated, the N–H ip mode shifts to low frequencies ($<1540 \text{ cm}^{-1}$) from 1571 cm^{-1} . These observations indicate that when one of the inner (B or C) rings is deprotonated, the N–H ip mode downshifts, which is consistent with the Osoegawa's paper. In our Raman data, a 1565 cm^{-1} peak appears in the first spectrum (ESA1, Fig. 2), indicating that four of the pyrrole N are protonated. We added sentences below in line 205-210, p. 7, and two references in line 200, p. 7.

“We also performed QM/MM calculation for the model I BV (Fig. S2) with four protonated pyrrole nitrogens and with one deprotonated pyrrole nitrogen in ring B or C (Fig. S4). The calculation shows that N–H ip mode appears at 1571 cm^{-1} with the four protonated nitrogens (top panel, Fig. S4), while the N–H ip mode downshifts to $<1540 \text{ cm}^{-1}$ with deprotonation of B or C ring (middle and bottom panels, Fig. S4). The calculation supports our signal assignment of the 1564 cm^{-1} peak to the N–H ip mode.”

Figure S4. Calculated Raman spectra of BV with protonated and deprotonated pyrrole rings. QM/MM calculation of Raman spectra of miRFP670/C20A (model I in Fig. S2) for (top) BV with four protonated pyrrole nitrogens, (middle) BV with one deprotonated pyrrole nitrogen in ring C, (bottom) BV with one deprotonated pyrrole nitrogen in ring B. Black and red lines indicate spectra in H₂O and D₂O, respectively.

Figure 3:

The model includes two kinds of the bound biliverdin.

I wondered how these two are deduced from their experiment.

These two biliverdin species were defined earlier in the C20S mutant of BphP1-FP, which is a dimeric form of miRFP670 (ref 20, Shcherbakova *et al.*, *Chem. Biol.* 22, 1540 (2015).). We added the following explanation in line 99-103, p.4;

“In BphP1-FP, which is a dimeric protein engineered from the same parental protein as miRFP670 (*RpBphP1*)²¹, C–S binding to Cys in the GAF domain both via C3¹ and C3² were detected in its 3D structure. Therefore, we assume that C–S binding of BV to the GAF domain in miRFP670/C20A occurs via C3¹ and C3² as well.”

In our Raman spectra, we could not distinguish the C-S binding to the GAF domain via C3¹ or C3².

Figure 4:

Peak positions should be written in c,d.

We added indicators.

Line 433

H254 and H284 → His254 and His284 (for consistency in the main text)

We changed these.

Reviewer #2 (Remarks to the Author):

This is an interesting paper that investigates the binding of the BV chromophore to the apoprotein of a near-infrared fluorescent protein (miRFP670) developed from a bacteriophytochrome. In particular, using time-resolved stimulated Raman spectroscopy techniques, the paper presents real-time data that offer new insights into this process, which are likely to be useful in current efforts to increase the fluorescence quantum yields of such proteins and thereby facilitate their application as bioimaging tools. Specifically, the paper provides valuable details of how the binding of the BV chromophore to two cysteine residues (one in the PAS domain and one in the GAF domain) of miRFP670 occurs. Given that this dual binding (through two thioether bonds) is a key feature separating miRFP670 from very weakly fluorescent native bacteriophytochromes, which bind the BV chromophore through a single cysteine residue, it has been argued that the increased rigidity offered by the additional thioether bond plays an important role in increasing the fluorescence quantum yield of miRFP670. Accordingly, it is worthwhile to gain a better understanding of the binding of the BV chromophore to the two cysteine residues, as provided by the present paper.

We thank the reviewer for the positive feedback and appreciation of the importance of the research.

Still, I think there are two issues that need to be addressed in a revised manuscript before this paper can be considered for publication in Communications Chemistry.

(1) First, while the paper considers the native miRFP670 protein and a mutant lacking the cysteine residue in the PAS domain, it seems to me that a better appreciation of the problem at hand would also require comparison with a mutant lacking the cysteine residue in the GAF domain.

We agree that BV binding dynamics in a mutant that has a Cys only in the PAS domain would be useful. Note that such a 'mutant' would actually represent the situation in native BphPs, as in BphPs BV binds to a Cys in the PAS domain only. We did not generate a miRFP670 mutant that lacks the Cys in PAS, but we used the nearly identical protein miRFP709, which derives from the same template as miRFP670, RpBphP1 from *R. palustris*, and has a native Cys in the PAS domain in the same position as miRFP670, but not in the GAF domain. Besides the presence/absence of a Cys in the PAS domain, miRFP670 and miRFP709 differ only by 2 amino acids. Accordingly, we performed time-resolved absorption and Raman experiments in miRFP709. We found an evolution similar to that of the early stages of the miRFP670 C20A mutant. Covalent binding to the Cys in the PAS domain occurred with a time constant of 18 min, which is similarly slow as that determined for miRFP670 (14 min). Thus, we conclude that covalent binding to Cys in PAS is an intrinsically slow process.

We added data in **Figs. 5 and S11**, and a paragraph in line 340-376, pp. 11-12.

"Time-resolved experiments on miRFP709

To assess the binding dynamics in miRFPs in the presence of a Cys in the PAS domain only, and not in the GAF domain, we further studied BV incorporation dynamics in miRFP709. miRFP709 is

derived from the same template as miRFP670, RpBphP1, and apart from the presence/absence of a Cys in the GAF domain, it differs from miRFP670 by only two amino acids.²⁹ The position of the Cys in the PAS domain is identical to that of miRFP670 and native BphPs. Consequently, its absorption/emission spectra peak at 683 and 709 nm respectively, similar to that of native BphPs and significantly red-shifted with respect to that of miRFP670. **Fig. 5a** shows selected time-resolved absorption spectra up to 60 min after adding apoprotein of miRFP709 to BV solution. The time-resolved absorption spectra were globally fitted with three time components: 2.1 min, 18 min, and an infinite component (**Fig. 5b**). The Soret-band absorption intensity dropped by ~40% immediately after adding the apoprotein (in <1 min). Also, the Q-band absorption rose up in <1 min, peaking at 674 nm (black line, **Fig. 5b**). In 2.1 min, the Q-band absorption rose up by ~30% with a 2-nm peak shift to 676 nm (from black to red, **Fig. 5b**). In 18 min, the Q-band absorption rose up more (by ~40%) with a peak shift to 683 nm (from red to blue, **Fig. 5b**), which did not evolve any more in the experimental temporal range.

Figs. 5c-e and **S11** show globally fitted time-resolved stimulated Raman spectra of miRFP709 after adding the apoprotein to BV solution, fitted with the same time constants as the time-resolved absorption spectra in **Fig. 5b**. Immediately after the apoprotein mixing, a peak at 1561 cm^{-1} increased, similar to miRFP670 (**Fig. 4e**) and its C20A mutant (**Fig. 2f**), which is assigned to an N–H in plane (ip) bending mode, a marker of BV protonation (from gray to black, **Fig. 5e**). We assign this fast reaction within 1 min to BV isomerization, protonation, and localization to the chromophore binding pocket of miRFP709, as we suggest in miRFP670 and its C20A mutant. In 2.1 min, the Raman signals rose in the entire spectral region (from black to red, **Fig. 5c-e**), resulting in a similar Raman spectrum to that of miRFP670 and its C20A mutant (2nd EAS, red lines in **Figs. 2f, 4e and 5e**). The Raman spectral similarity of the 2nd EAS and the similar time constant (1.5 min and 2.7 min for miRFP670 and its C20A mutant, respectively) imply that the reaction in 2.1 min is hydrogen bond formation/rearrangement, as in miRFP670 and its C20A mutant. In 18 min, the Raman signal intensities further increased (from red to blue, **Fig. 5c-e**), consistent with the increase of the absorption. Also Raman spectral shifts were detected at $\sim 1600\text{--}1650\text{ cm}^{-1}$ in 18 min, indicating that the C=C backbone structure changed. We assign the 18-min reaction to covalent bond formation to Cys in the PAS domain. We note that such relatively slow binding dynamics to Cys in PAS (18 min) are consistent with the slow process in miRFP670 (14 min), which was assigned to Cys in PAS binding as well. We conclude that covalent binding to Cys in PAS is an intrinsically slow process. In **Fig. 5f**, the BV binding model of miRFP709 is shown.”

Figure 5. Time-resolved absorption and Raman spectra of miRFP709 at pH 8.0. (a) Selected time-resolved absorption spectra. (b) Evolution-associated spectra (EAS) of globally fitted time-resolved absorption spectra. (c-e) Evolution-associated spectra (EAS) of globally fitted time-resolved stimulated Raman spectra with an 800 nm Raman pump. The indicated time constants denote the lifetime by which each EAS evolves into the next one. (f) BV incorporation model of miRFP709.

(2) I think parts of the presentation contain perhaps too much “spectroscopic jargon” and are a little bit too vague to be fully appreciated by a broad readership. This holds true especially for the Section “Time-resolved experiments in the miRFP670/C20A mutant”. For example, on page 6, these points are illustrated by the sentences:

“As shown in Fig. 2b, the Q-band intensity was increased by 2.5-fold. Thus, the ~2-fold signal increases such as at ~700 and 974 cm⁻¹ are considered as a result of an increase of the pre-resonance with the 800 nm Raman pump. However, the appearance of some peaks such as at 818, 1275 and 1317 cm⁻¹ cannot be explained only by the higher resonance effect; thus, these peaks are derived from BV conformational change in <1 min.”

We believe that the ‘vagueness’ in the paragraph mentioned by the reviewer stems from simultaneous discussion of absorption and Raman spectra. To clarify the text, we specified whether a statement concerns Raman or absorbance and made the description more specific in the paragraph (line 148-168, p. 6)

“Immediately after adding the apoprotein of miRFP670/C20A to BV (<1 min component, evolution from gray to black lines, **Fig. 2d–f**), a substantial **Raman** signal increase was observed at 818, 1275, 1317, 1565, 1626 and 1647 cm⁻¹. Generally, the pre-resonance Raman intensity is dependent on the intensity and spectrum of an absorption band close to the Raman pump, in this case the Q-band absorption (**at ~670 nm**). As shown in **Fig. 2b**, the Q-band **absorbance** increased by 2.5-fold **within <1 min (from gray to black lines)**. Thus, the ~2-fold Raman signal increase **immediately after the sample mixing (from gray to black lines, Fig. 2d–f)**, such as at ~700 and 974 cm⁻¹, is considered as a result of an increase of the pre-resonance with the 800 nm Raman pump. However, the appearance of some particular **Raman** peaks such as at 818, 1275 and 1317 cm⁻¹ cannot be explained only by the increased resonance effect. Thus, these peaks result from BV conformational changes in <1 min.

In 2.7 min (evolution from black to red lines, **Fig. 2d–f**), strong peaks appeared at 679 and 713 cm⁻¹, in addition to ~2-3-fold peak increases at 818, 974, 1275, 1317, 1565, 1626 and 1649 cm⁻¹. Especially, the signals around 679 cm⁻¹ increased more than 4-fold. Therefore, the 679 cm⁻¹ signal rise probably results from a structural change of BV in 2.7 min.

In 19 min (evolution from red to blue lines, **Fig. 2d–f**), a substantial **Raman** signal transition was observed in the C=C stretching region: the 1626-cm⁻¹ peak is upshifted to 1630 cm⁻¹ with a ~20% signal loss, while the 1647-cm⁻¹ band intensity rose by ~15% without a detectable peak shift. Overall, the Raman signal became weaker in 19 min, which probably resulted from a decreased resonance with the 800 nm Raman pump associated with the absorption blue-shift from 671 to 643 nm (**Fig. 2b**).”

REVIEWERS' COMMENTS:

Reviewer #1 (Remarks to the Author):

The authors satisfactorily replied to my concerns and revised the manuscript correspondingly.

Minor points

Line 58: To fluorescence => To fluoresce

Line 61: GFP-like family that absorb and fluorescence ...
=>GFP-like family that absorbs and fluoresces ...

Line 225: the Raman intensity of rose.. => the Raman intensity of ?? rose ...

Line 270: 1650 cm⁻¹ (-1 should be superscript)

Line 336: Pas => PAS

Figure S1: The heights of right (a, b) and left (c, d) plots are not consistent.

Figure S3, caption: Figure 2d-f and S2 => not bold (?)

Figure S5: The heights of right and left plots are not consistent. The caption should explain about what are right and left plots.

Figure S6 and S9: Figure S6 shows EAS3-EAS2, but Figure S9 shows EAS2-EAS3. They are not easy to compare.

Figure S10: The title of figure and the caption should be written separately, as in Figure S9.

Reviewer #2 (Remarks to the Author):

I am happy to recommend the revised version of this manuscript for publication in Communications Chemistry.

We appreciate the reviewers' careful consideration of the manuscript and have made amendments accordingly.

Reviewer 1:

Minor points

Line 58: To fluorescence => To fluoresce

corrected

Line 61: GFP-like family that absorb and fluorescence ...

=>GFP-like family that absorbs and fluoresces ...

corrected

Line 225: the Raman intensity of rose.. => the Raman intensity of ?? rose ...

We clarified this by writing 'the Raman intensity in the entire spectral region rose...'

Line 270: 1650 cm-1 (-1 should be superscript)

corrected

Line 336: Pas => PAS

corrected

Figure S1: The heights of right (a, b) and left (c, d) plots are not consistent.

corrected

Figure S3, caption: Figure 2d-f and S2 => not bold (?)

corrected

Figure S5: The heights of right and left plots are not consistent. The caption should explain about what are right and left plots.

corrected

Figure S6 and S9: Figure S6 shows EAS3-EAS2, but Figure S9 shows EAS2-EAS3. They are not easy to compare.

We thank the reviewer for pointing this out, Figure S9 has been changed to EAS3-EAS2 and EAS4-EAS3.

Figure S10: The title of figure and the caption should be written separately, as in Figure S9.

We changed the panel order of Fig. S10 and its caption accordingly.